# An anionic two dimensional covalent organic framework from tetratopic borate centres pillared by lithium ions

Darosch Asgari [1], Julia Grüneberg[1], Yunkai Luo[2], Hüseyin Küçükkeçeci[1], Samrat Ghosh[1,3], Veniamin Chevelkov[4], Sabrina Fischer-Lang[5], Jérôme Roeser[1], Adam Lange [4], Bruce Dunn [2], Michael Gradzielski [1] & Arne Thomas [1] ✉

Non-covalent interactions play an important role for the framework formation of two-dimensional covalent organic frameworks. Until now, π–π interactions and hydrogen bonding are the main reported forces facilitating the stacking of framework layers. Here, we present a two-dimensional anionic covalent organic framework based on tetratopic borate linkages, where layers are connected by ionic interactions between the linkage site and counter cations. The crystalline covalent organic framework is accessed through the formation of an amorphous borate-based polymer and subsequent solvothermal treatment. The progress of crystallization is investigated, revealing the crystallite growth and morphological change from agglomerated dense particles to hollow crystallite spheres. Due to the pillared nature, the crystallites can be exfoliated into nanosheets by sonication of the material in the presence of methanol. The crystallization and ordered arrangement of the lithium ions in the interlayer space is shown to benefit the conductivity tenfold compared to the amorphous material.

Two-dimensional covalent organic frameworks (COFs) are crystalline materials that have emerged as an attractive platform for solid-state conduction and electrochemical applications due to their intrinsic porosity, robust covalent bonds, low density and high functional tunability[1,2]. While the growth of the individual framework layer is achieved by the formation of covalent bonds, non-covalent forces such as π–π interactions and hydrogen bonding are of equal relevance for the framework formation, as the modulation of the stacked layers can drastically influence the crystallinity and properties of the material[3–6]. Regarding the design of COFs for lithium-ion conduction, extrinsically and intrinsically lithium-ion conductive COFs can be differentiated[7]. For extrinsically lithium-ion conductive COFs, the porous nature of the material is used solely to load external lithium ions as dissolved salts into the framework. Intrinsically lithium-ion conductive COFs; however, use anionic structural motifs located in the skeleton of the framework, which induce the incorporation of extra-framework cations to offset the framework charge. Here, the counter cation is an equally integral part of the framework due to the necessity to achieve charge neutrality. However, while the incorporation of the anionic building blocks is well described, the location and coordination of the respective counter cations is often unclear. It is usually assumed that the countercations of the anionic framework are located randomly within the pores[8]. For two-dimensional frameworks, the ionic interactions of the charged layer and counter ion could serve as an alternative non-covalent interaction. Instead of occupying the pore space, the interlayer space would be utilized as a coordination site for the counter ions not impacting the pore space. Additionally, the interlayer space would offer regular channels, which could be beneficial for ionic conductivity. Tetrahedral

[1]Institute of Chemistry, Technische Universität Berlin, Berlin 10623, Germany. [2]Department of Materials Science and Engineering, University of California, Los Angeles, CA 90095, USA. [3]Inorganic and Physical Chemistry Laboratory, Council of Scientific and Industrial Research (CSIR), Central Leather Research Institute (CLRI), Chennai 600020, India. [4]Research Unit Molecular Biophysics, Leibniz-Forschungsinstitut für Molekulare Pharmakologie, Berlin 13125, Germany. [5]Organic Materials Chemistry, Osnabrueck University, Osnabrueck 49069, Germany. ✉e-mail: arne.thomas@tu-berlin.de

borate centres ($[BO_4]^-$) offer access to an anionic building block alongside good thermal stability and low relative weight. This results in a high wt% ratio for the respective lithium counter ions, which makes it an attractive motif for solid-state conduction. So far, the synthesis of borate-based COFs was achieved using $B(OMe)_3$ in combination with a suitable base that can supply the countercation. Due to the in situ generated methanolate, only chelating linkers are tolerated, making only anionic spiroborate centres accessible[9–11]. Unfortunately, this impedes the use of monodentate hydroxy and phenol-bearing linkers, as they would result in tetratopic borate linkages, which are susceptible toward the alcoholate generated.

We report a simple synthetic strategy, that enables the construction of an anionic COF from tetratopic borate centres using 4,4′-biphenol as linker. In our approach, first, an amorphous network is formed and subsequently crystallized[12]. An unprecedented structure was found for the obtained anionic COF, as the anionic framework layers are pillared by ionic interactions with the lithium counter ion. Here, the lithium ions are coordinated in the interlayer space to facilitate effective charge compensation.

## Results and discussion

The synthesis of a biphenyl-linked borate-based covalent organic framework (BPB-COF) was achieved in a two-step process from lithium borohydride as simultaneous boron and lithium source with 4,4′-biphenol as linker. First, the phenol groups of the linker are deprotonated irreversibly by lithium borohydride, releasing hydrogen gas as a by-product for the formation of a biphenyl-linked borate polymer (BPB-Poly). Since the polymerization occurs in a short time span, first, a kinetically controlled amorphous polymer network is generated (BPB-Poly, Fig. 1b). Following the concept of dynamic covalent chemistry[13], a transformation to the crystalline compound requires the cleavage and reformation of the B−O bonds present, enabling the conversion of the kinetic and amorphous material to the thermodynamically favoured crystalline state. This can be achieved through elongated reaction times, high pressure, and high temperatures. Therefore, the resulting polymer-solvent mixture was transferred to a Teflon-lined steel autoclave and 1 equivalent triethylamine was added to the suspension in order to deprotonate any residual phenol groups and assist as a competing nucleophile in the cleavage and reformation of the B−O bond. The autoclave was sealed shut and placed in the oven at 150 °C for at least 2 days (Supplementary Table 1). After cooling, the resulting white precipitate was filtered under an inert atmosphere, washed with dry tetrahydrofuran (THF) and dried at 100 °C under vacuum to yield BPB-COF (Fig. 1b). Due to the susceptibility of BPB-COF towards moisture it was handled and stored under inert atmosphere (Supplementary Fig. 6). For characterization purposes lithium

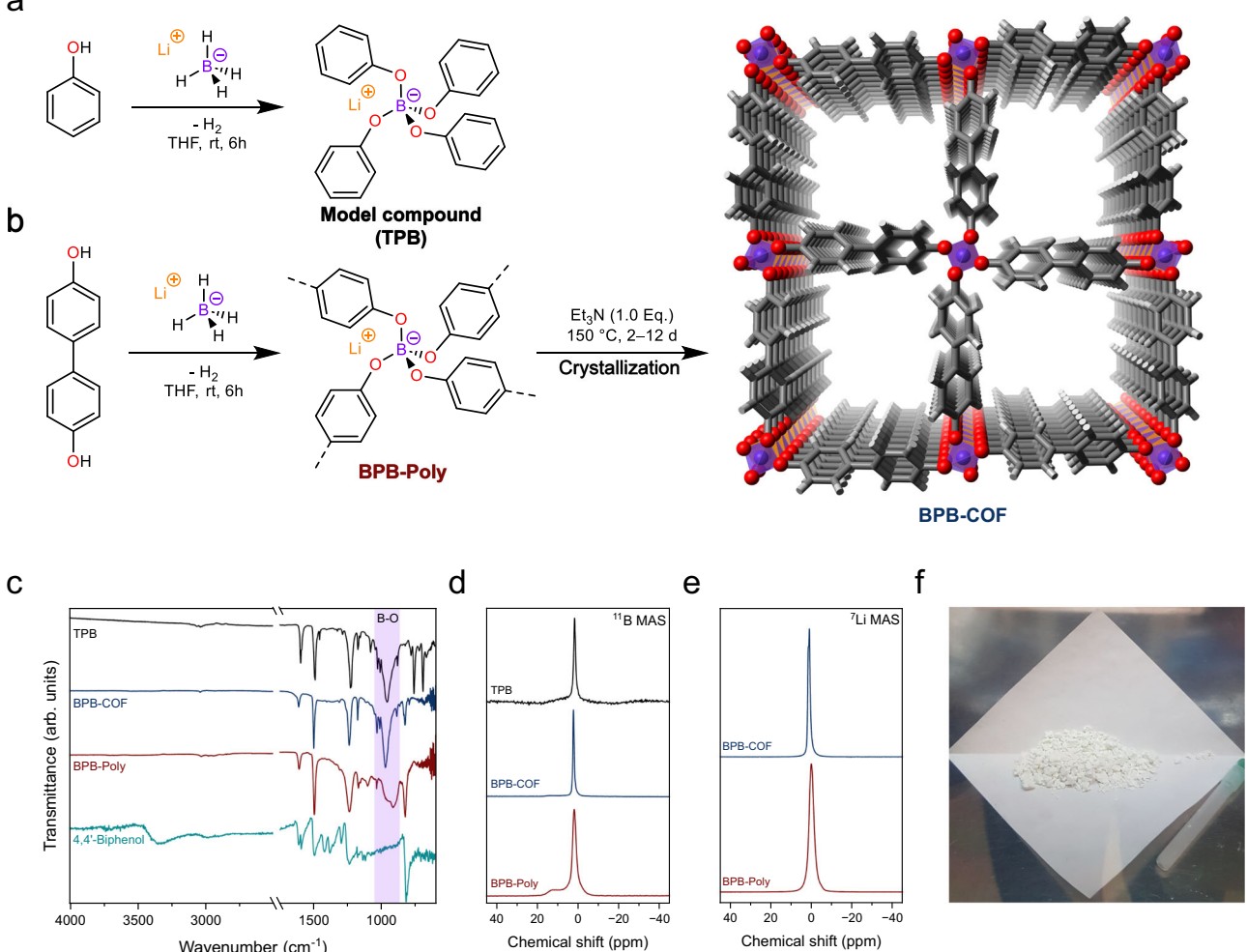

**Fig. 1 | Synthetic strategy and compositional analysis. a** Synthesis of tetra-phenoxyborate (TPB) as a model compound. **b** Synthesis of BPB-Poly and conversion to BPB-COF. **c** FT-IR spectra of BPB-COF (blue) and BPB-Poly (red) along with the model compound TPB (black) and linker (cyan). **d** $^{11}$B MAS NMR of the model compound TPB (black), BPB-COF (blue), and BPB-Poly (red). **e** $^7$Li MAS NMR showing the presence of lithium ions offsetting the anionic net charges. **f** One-batch synthesis yielding 1.2 g BPB-COF.

tetraphenoxyborate (TPB) was synthesized from lithium borohydride and phenol as model compound (Fig. 1a).

In the Fourier-transform infrared (FT-IR) spectrum, the B–O stretching mode at 963 cm$^{-1}$ shows the successful linkage formation, together with the absence of the O–H stretching band at 3359 cm$^{-1}$, indicating the full conversion of the 4,4'-biphenol (Fig. 1c). The incorporation of 4,4'-biphenol in both networks is confirmed through $^{13}$C cross-polarization magic-angle spinning nuclear magnetic resonance (CP-MAS NMR) (Supplementary Fig. 1). For BPB-Poly, residual THF can be seen within the network in contrast to BPB-COF, which is absent of a $^{13}$C signal that can be assigned to THF (Supplementary Fig. 1)[14].

To verify the presence of the tetratopic [BO$_4$]$^-$ species, the materials were investigated by recording directly excited $^{11}$B MAS NMR spectra (Fig. 1d). In order to confirm the tetratopic borate species of BPB-Poly and BPB-COF, the chemical shift was compared to the molecular tetraphenoxyborate species of the model compound TPB. The tetraphenoxyborate species was found at a chemical shift of 1.70 ppm, which is consistent with values for molecular tetratopic borates found in literature[15]. The $^{11}$B MAS NMR spectra of BPB-Poly were in accordance with a resonance signal at 1.87 ppm. In addition to the desired signal of the tetrahedral [BO$_4$]$^-$ species, the quadrupolar pattern signal can be attributed to trigonal [BO$_3$] in the range of 0–20 ppm, evidencing the presence of defects within BPB-Poly[16]. After solvothermal treatment, the attenuation of the quadrupolar [BO$_3$] signal can be observed, with the signal for the desired [BO$_4$]$^-$ at $\delta = 2.3$ ppm (Fig. 1d). The change in chemical shift suggests that while both materials are constructed from the desired tetratopic borate nodes, the ligand to boron interactions are slightly altered in the crystalline phase.

The negative backbone charge of the networks is balanced by lithium counter ions, which can be evidenced through the directly excited $^7$Li MAS NMR spectrum. The $^7$Li resonance at 0.1 ppm observed for BPB-Poly is typical for solvated lithium ions, due to residual THF present within the network. Upon crystallization, the $^7$Li resonance signal of BPB-COF is found to shift to 1.09 ppm, evidencing a changed environment for the lithium ions (Fig. 1e). The quantitative amounts of boron and lithium determined through inductively coupled plasma optical emission spectrometry (ICP-OES) are in accordance with the theoretical amount. These are, to our knowledge, the highest boron and lithium contents reported for borate-based COFs (Calc. B, 2.80; Li, 1.80. Found: B, 2.78 ± 0.12; Li, 1.64 ± 0.07, Supplementary Fig. 24 and Supplementary Table 2).

While for BPB-Poly no long-range order is detected through powder x-ray diffraction (PXRD) analysis, through solvothermal treatment, the polymer could be successfully converted to the crystalline BPB-COF (Fig. 2a). By prolonging the duration of the treatment an increase in crystallinity is observed in the diffractogram through the appearance of additional diffraction peaks at low $2\theta$ values combined with a reduced full width at half maximum (FWHM), indicative of an increase in crystallite size (Fig. 2a)[17].

After the subtraction of the background (Supplementary Fig. 2), the PXRD pattern was indexed on a primitive tetragonal unit cell (P-lattice) with lattice constants of $a = b = 16.7557$ Å and $c = 5.0126$ Å. As the anionic framework is based on tetrahedral [BO$_4$]$^-$ sites reticulated by linear biphenyl units, the formation of a three-dimensional (3D) net with diamond (**dia**) topology was assumed, as the **dia**-net is a regular 4-connected (4-c) net based on tetrahedral nodes. Therefore, we assumed BPB-COF should crystallize in a single (**dia**) or, most likely, intergrown diamond net (**dia-c**N, with N defined as the number of interpenetrating net components)[18]. However, no feasible result could be derived from the general formula that establishes the interpenetration degree of **dia**-nets (Supplementary Fig. 3)[19]. Moreover, fitting **dia**-nets with various interpenetration degrees within the indexed cell parameters was in poor accordance with the experimental

PXRD pattern (Supplementary Fig. 5) due to the absence of the characteristic diffractions at $2\theta = 14.9$ and 17.6, leading to the investigation of other topologies. Interestingly, a two-dimensional square-lattice (**sql**) net was found to fit well within the indexed cell parameters and was in very good agreement with the experimental pattern (Fig. 2b, Supplementary Figs. 4 and 5). Both **sql** and **dia** are regular nets based on 4-c vertices, however, while the **dia** topology is based on tetrahedral vertices, the **sql**-net is constructed from planar rectangular building units resulting in a 2-periodic net. So far, COFs with a **sql** topology have been constructed from 4-coordinated rectangular building-units. Although the tetratopic borate is a tetrahedral building block, the **sql** topology is enabled due to the flexibility of the B–O–C linkages around the borate unit, allowing the 4 points of extensions of the node to be coplanar and thus form such a layered structure (Fig. 2d).

As two-dimensional COFs are based on 2-periodic nets, noncovalent interactions such as π–π interactions and hydrogen bonding assist in the construction of the framework through stacking of the layers. For our **sql**-based model, this is unrealistic, due to the torsion between the phenyl rings and the biphenyl linker oriented diagonally in the cell. The interlayer distance of 5.016 Å is also found to be much larger than common values for two-dimensional COFs and outside the range for significant contributions through π–π interactions[3,5,20]. Furthermore, the negative charge of the borates and the additional electrostatic repulsion of the layers caused by the anionic net charge of the framework further refutes the possibility of layers stacked by π–π interactions. Instead, a pillaring of the layers is driven by electrostatic interactions between the lithium counter ion located in the interlayer space and the phenoxy moieties from two different layers. The coordination is based on the formation of LiO$_4$ tetrahedra leading to the effective charge compensation of the framework (Fig. 2d). To further validate the environment of the lithium ions present in BPB-COF, we conducted first principles quantum mechanical NMR calculations to determine the theoretical chemical shielding. Using CASTEP, the isotropic chemical shift for the lithium-ion of BPB-COF was calculated to be 1.19 ppm (Supplementary Fig. 25)[21,22]. This is in accordance with the experimentally measured value of 1.09 ppm further supporting the tetrahedral coordination in the interlayer space. It is worth noting that the coordination of the counter-cation by the phenoxy groups of the borate was also observed in molecular tetraphenoxyborates and even found to act as structural support, linking molecular borate subunits to infinite one-dimensional polymeric structures similar to what is observed for the structure of BPB-COF[23,24]. Consequently, the lithium ions become structurally and topologically relevant, with the final topology being best described by a pillared **sql**-net (Fig. 2c). In a series of Rietveld refinements[25,26] where not only the $R_{wp}$ and $R_p$ values but also the potential energy of the structure was considered (applying the universal force field)[27] a final model with good $R_{wp}$ and $R_p$ values of 6.51% and 3.39% together with sensible bond lengths of O–Li = 1.978 Å and O–B = 1.493 Å could be obtained (Fig. 2e, final cell parameters: P-4, No. 81, $a = b = 16.729$ Å and $c = 5.016$ Å, boron (Wyckoff positions 1b and 1d), lithium (Wyckoff positions 1a and 1c))[28,29].

Since both borate-based materials displayed good thermal stability as evidenced by thermogravimetric analysis (TGA, Supporting Fig. 9), the porosity and sorption capacities were assessed on samples activated at 150 °C under high vacuum (Supplementary Fig. 7). While BPB-Poly shows no significant uptake of nitrogen (N$_2$), a remarkable increase in microporosity and N$_2$ uptake is found for the BPB-COFs (Fig. 2f). Using the BETSI tool[30], after 2, 5, and 12 days of solvothermal treatment, respective surface areas of 802, 784 and 819 m$^2$ g$^{-1}$ were calculated with a pore size distribution centering around 0.61–0.64 nm (Fig. 2g, Supplementary Figs. 10 and 11). Notably, while the elongation of the solvothermal treatment results in an improved crystallinity (Fig. 2a), no significant enhancement in surface area could be detected through N$_2$ sorption analysis (Fig. 2f, inset).

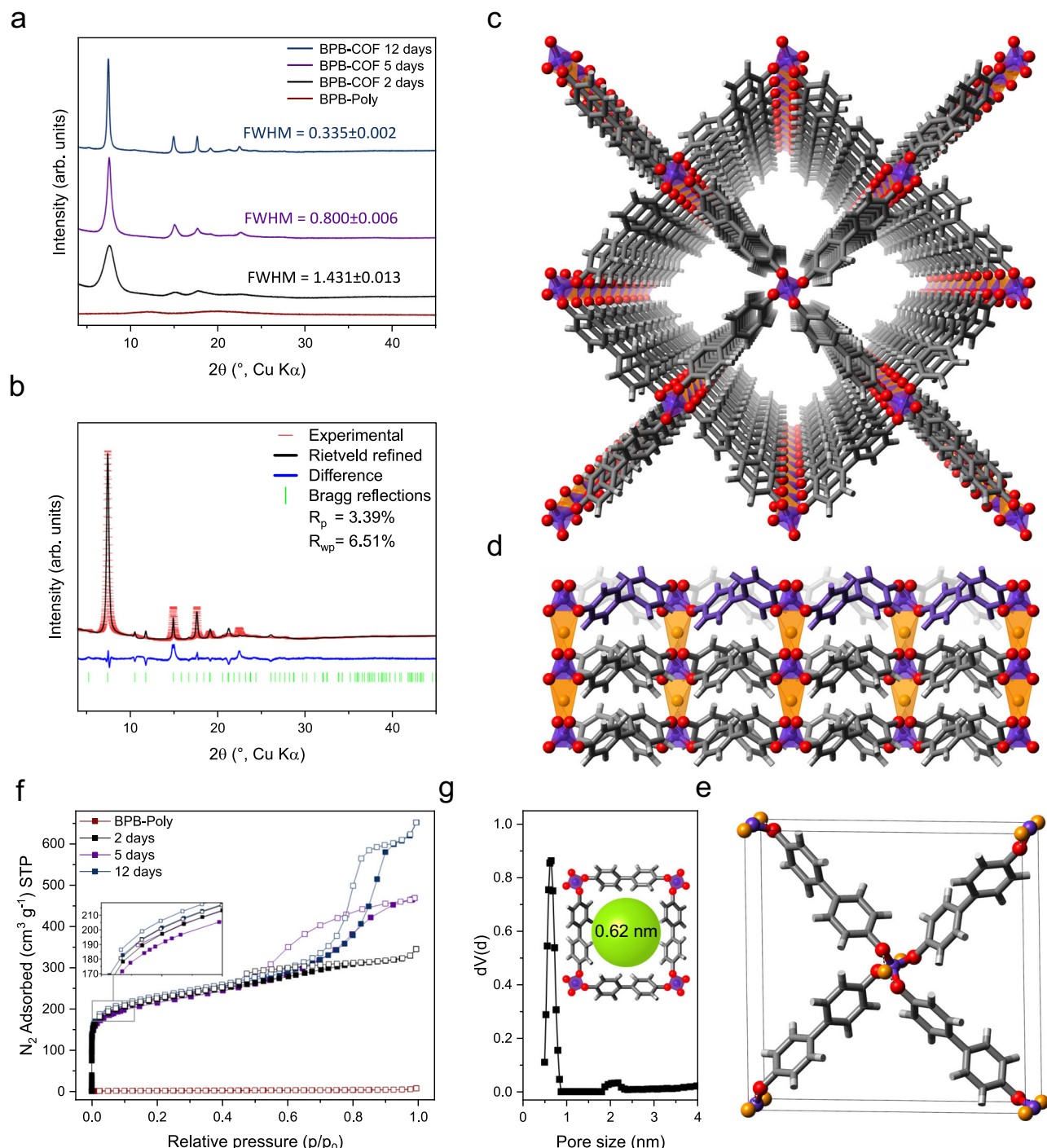

**Fig. 2 | Structural characterization of BPB-COF. a** Collected PXRD pattern for BPB-Poly (red) and BPB-COF after 2 (black), 5 (purple), and 12 (blue) days. **b** Experimental PXRD pattern (red), obtained profile after Rietveld refinement (grey), the difference between the experimental and refined profile (blue) and positions of observed diffractions (green). **c** Two-dimensional **sql** structure viewed from the *c*-plane. **d** Structure of BPB-COF viewed from the b-plane showing the interlayer lithium ions (yellow) enabling pillaring through ionic interactions and highlighted diagonal in-plane oriented linker. **e** Unit cell of BPB-COF. **f** Nitrogen sorption isotherms for the amorphous BPB-Poly (red) and crystalline BPB-COF after 2 (black), 5 (purple), and 12 (blue) days of solvothermal treatment showing a similar uptake at low-relative pressures. **g** Pore size distribution with inset depicting the pore and calculated pore size.

For partial pressures >0.5$p/p_0$ (Fig. 2f), an increasing hysteresis loop was observed for more crystalline samples. Through investigation with scanning and transmission electron microscopy (SEM and TEM), the emerging hysteresis was found to be consistent with crystallite length and can be attributed to the additional meso/macroporosity created through the interstices of the agglomerated crystallites (Fig. 3).

Starting with BPB-Poly, dense agglomerated spheres with an irregular surface are obtained (Fig. 3, 0 days, Supplementary Figs. 12 and 17). After 2 days of solvothermal treatment, the particles are found to retain their dense nature with a new uniform surface (Fig. 3, 2 days, Supplementary Figs. 13 and 18). Prolonging the solvothermal treatment led to continuous recrystallization of the material resulting in the hollowing of the spherical particles and growth

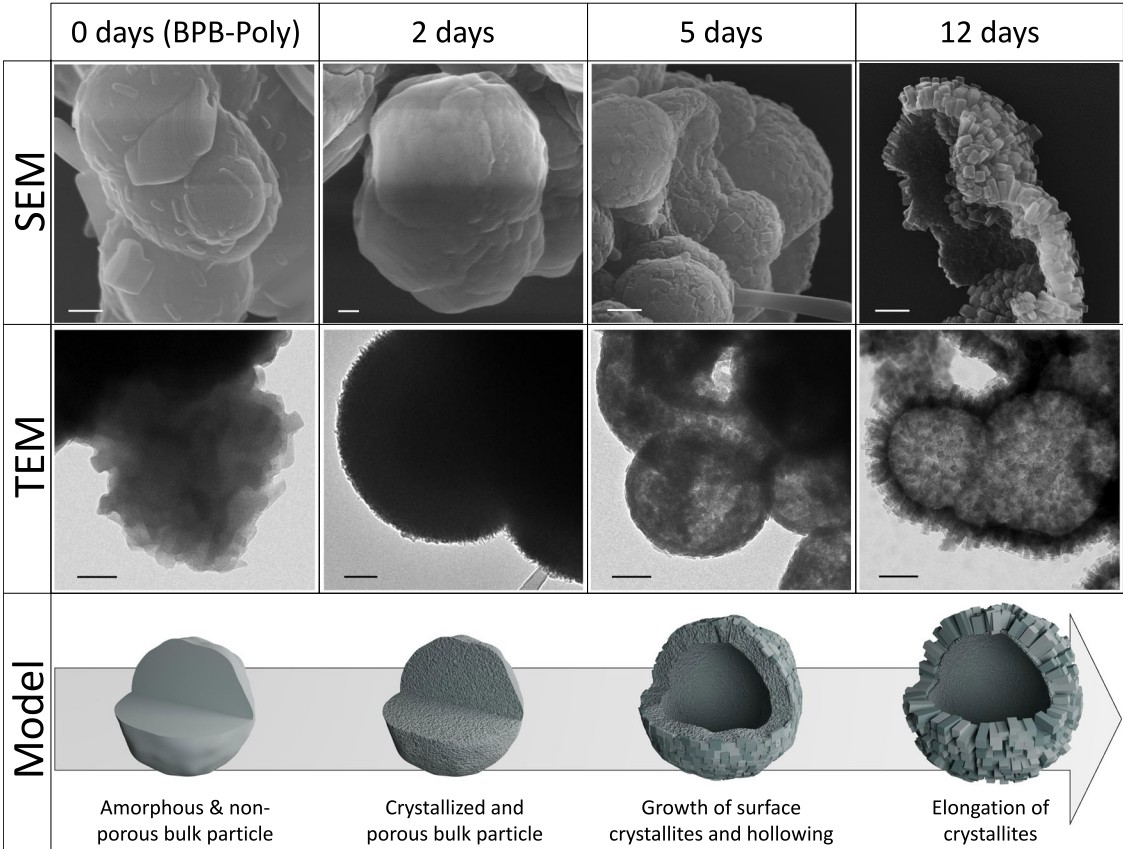

**Fig. 3 | Morphological change of the particles throughout the crystallization.** SEM images of BPB-Poly and BPB-COF after 2, 5, and 12 days of solvothermal treatment show the crystallization process of the bulk material and the growth of rod-shaped crystallites on the surface alongside a model depiction of the particles. For BPB-COF (12 days), a broken particle was chosen to highlight the hollow nature of the sphere. TEM images of BPB-Poly and BPB-COF (2, 5, and 12 days) show the conversion from dense and diffuse to hollow spherical particles with increasing surface-crystallite length. Scale bars set to 200 nm.

of crystallites on the surface (Fig. 3, 5 days, Supplementary Figs. 14 and 19). Further treatment resulted in the elongation of these crystallites, as the process is driven by thermodynamics to minimize the overall free energy of the system, with larger crystals having a minimized surface energy. Due to this process, the bulk phase is further dissolved to recrystallize on the already formed crystallites located at the particle surface. This led to the final morphology of hollow spheres covered with agglomerated rectangular cuboid crystallites in the 100–200 nm range (Fig. 3, 12 days, Supplementary Figs. 15 and 20).

Having the rectangular cuboid crystallites of BPB-COF, we hypothesized the exfoliation of crystallites in a top-down approach. As the layers are comprised of covalent bonds, disruption of the weaker ionic interactions pillaring the layers should yield borate-based covalent organic nanosheets (BPB-NS). We also hypothesized that in order to achieve a chemical exfoliation the solvent should have an affinity for the lithium ion in order to compete with the ionic interactions between the phenoxy moieties and the lithium ions. While ultrasonication in THF proved to be unsuccessful, instead the exfoliation could be achieved by using dry methanol. First, BPB-COF (12 days) was suspended in dry methanol, followed by sonication at 35 kHz for 30 minutes. After centrifugation to settle bigger particles and fragments, the desired BPB-NS were left natant in solution (Fig. 4a). For the analysis of the BPB-NS, the solution was simply dropped and cast onto the respective substrates. Through TEM, the obtained BPB-NS were found to have the expected dimensions of the prior observed rectangular cuboid crystallites of BPB-COF (12 days) in the approximate range of $50 \times 50$ nm (Fig. 4c and d, Supplementary Figs. 21 and 22). The thickness of the exfoliated layers was investigated through atomic force microscopy (AFM) in AC mode, where the BPB-NS were found to

consistently exfoliate < 1 nm thickness (Fig. 4b, Supplementary Fig. 22). Additionally, BPB-NS could be clearly distinguished in the AFM phase image due to the difference in roughness compared to the SiO$_2$ substrate (Fig. 4f).

Lastly, to determine the ionic conductivity of BPB-Poly and BPB-COF (12 days) electrochemical impedance spectroscopy was carried out on a tape-cast film. The tape-cast films were prepared under an argon atmosphere by mixing the active material powder and polyvinylidene fluoride in a 9:1 weight ratio in anhydrous N-methyl-2-pyrrolidone (Supplementary Fig. 23). After casting on a carbon-coated aluminium foil and drying, the respective thin-films were pouched and assembled in a coin cell, sandwiched between two stainless steel spacers. An anhydrous propylene carbonate solvent was dropped onto both thin films of BPB-Poly and BPB-COF electrodes for activation purposes before being assembled into a coin cell. Using ambient Nyquist Plots, the fitted $R_1$ value (Fig. 5a) represented the resistivity and was used to calculate the ionic conductivity. The ionic conductivity of BPB-Poly is $3.6 \pm 0.5 \times 10^{-6}$ S cm$^{-1}$, while BPB-COF displayed an approximately 10-fold increase in conductivity of $3.1 \pm 0.3 \times 10^{-5}$ S cm$^{-1}$ (Fig. 5b). The ionic conductivity values of both materials were found to increase with temperature from 20 to 60 °C, which indicated the hopping mechanism of lithium-ion transport in the materials. Moreover, the activation energy of lithium-ion transport in BPB-Poly was higher compared to BPB-COF (Fig. 5c), which indicates that the crystalline framework structure offers a regulated diffusion channel that lowers the hopping energy compared to the amorphous network underlining the beneficial influence of crystallinity on conductivity.

In conclusion, we have developed a strategy that allows the construction of a charged covalent organic framework from tetratopic

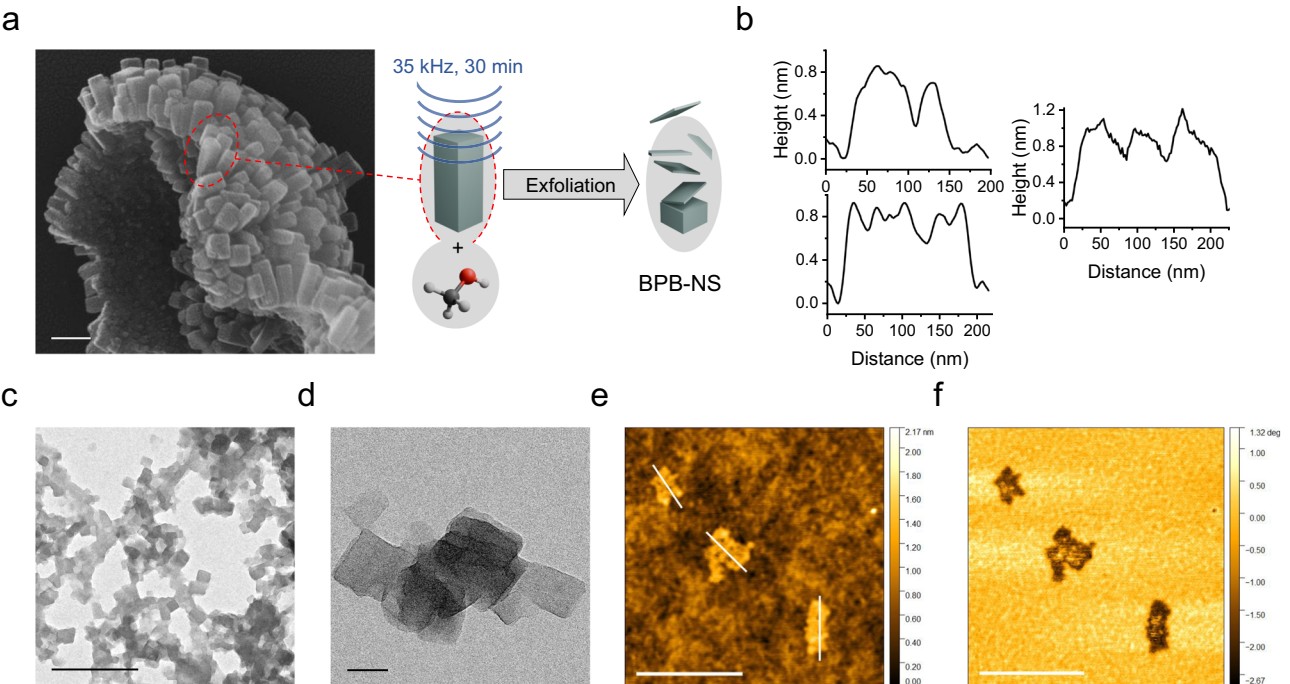

**Fig. 4 | Exfoliation and characterization of nanosheets. a** SEM image of BPB-COF (12 days) highlighting the crystallites and model of the exfoliation, the scale bar is 100 nm. **b** Height profile measured by AFM along the three lines shown in Fig. 4e. **c** TEM image giving an overview of BPB-NS, scale bar is 500 nm. **d** Close-up of layered BPB-NS, scale bar is 50 nm. **e** AFM topography image of exfoliated BPB-NS and marked height profiles, scale bar is 400 nm. **f** AFM phase image with BPB-NS clearly distinguished through surface roughness, scale bar is 400 nm.

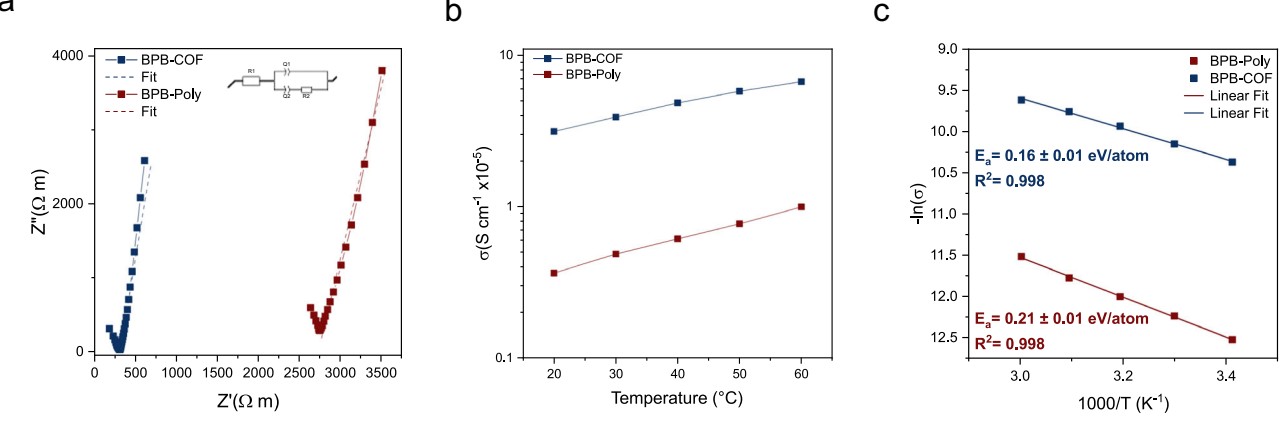

**Fig. 5 | Electrochemical measurements comparing BPB-Poly and BPB-COF. a** Room-temperature Nyquist plot (normalized by the ration of the area to thickness with the equivalent circuit model. **b** Temperature-dependent ionic conductivity and **c** Arrhenius character and activation energy of BPB-Poly and BPB-COF.

borate nodes connected by biphenyl units using lithium borohydride and 4,4'-biphenol as reactants. The synthesis was achieved in a two-step process of polymerization and crystallization. The two-dimensional framework was found to crystallize in a **sql**-net not relying on conventional π−π stacking but pillaring enabled by the counter-cation connecting the anionic framework layers through ionic interactions. The crystallization process was investigated by PXRD, SEM and TEM, showing the transformation from dense spheres to a microporous crystalline material with hollow spheres of agglomerated rod-shaped crystallites. The crystallites were exfoliated into <1 nm high rectangular nanosheets by sonication in methanol. Through EIS the influence of crystallinity on lithium-ion conductivity was investigated, showing a conductivity increase by approximately 10-fold when compared to the amorphous material.

## Methods
### General
Glassware was pre-cleaned with appropriate solvents and washed in a dishwasher employing *neodisher LaboClean A8* and deionized water. Unless otherwise noted, all reactions were carried out in oven-dried (120 °C) glassware. Fritted glass funnels (POR 4) used for filtration were cleaned with piranha-solution ($H_2SO_4$:$H_2O_2$ = 3:1) and flushed with deionized water. Magnetic stir bars were cleaned, depending on the amount and type of impurities, either with acetone and deionized water or in an $H_2SO_4$-bath and rinsed with deionized water. PTFE lined autoclaves were cleaned by using nitric acid (5 mL) at 120 °C overnight washing with deionized water and oven-dried at 120 °C prior to use. Reactions requiring inert conditions were either carried out in a glovebox (MBraun UNIlab pro, $O_2$ < 20 ppm, $H_2O$ < 1 ppm) or using

standard Schlenk technique employing Argon as inert gas. Moisture and air sensitive compounds were stored in an Argon-filled glovebox (MBraun UNIlab pro model plus). Unless otherwise noted, the employed chemicals were used without any further purification. Acetone and nitric acid (≥65% in dH₂O) were purchased from Roth. Anhydrous Tetrahydrofuran (THF) and anhydrous methanol (MeOH) was purchased from Thermo Scientific. Lithium borohydride (2 M in anhyd. THF) and triethyl amine, anhydr. was purchased from Sigma-Aldrich. Phenol was purchased from Alfa Aesar. 4,4′-Biphenol was purchased form TCI Chemicals.

### Solvothermal treatment

Solvothermal treatment was performed in PTFE-lined steel autoclaves, with preparation and work-up performed in an argon-filled MBraun glovebox type MB 120 BG. Borate networks were synthesized using either 45 or 125 mL autoclave (Parr Instrument Company, Model 4744). The PTFE liners were oven-dried overnight at 120 °C prior to use.

### Lithium tetraphenoxyborate (TPB)

An oven-dried Schlenk tube equipped with a stirrer and septum was charged with phenol (0.753 g, 8.00 mmol, 4.00 eq.), evacuated, and backfilled with Argon for a total of three times. Phenol was dissolved by the addition of anhydrous 1,4-dioxane (30 ml). The reaction was started by the slow addition of LiBH₄ (2 M in anhydrous THF, 1 ml, 2 mmol, 1.0 eq.). The reaction mixture was stirred for 30 min at room temperature and then heated to 70 °C and stirred overnight. After cooling to room temperature, the white solid was collected via Schlenk filtration and washed with anhydrous dioxane (10 ml), anhydrous acetone (10 ml) and anhydrous pentane (10 ml). The solid was dried under reduced pressure at 40 °C overnight. TPB was obtained as a white solid (92%, 0.720 g, 1.85 mmol). $^{11}$B-NMR (64 MHz, CDCN): $\delta$ = 2.52 ppm. $^{11}$B-NMR (CP-MAS): $\delta$ = 1.77 ppm. $^{7}$Li-NMR (78 MHz, CDCN): $\delta$ = −1.53 ppm. $^{1}$H-NMR (400 MHz, CDCN): $\delta$ = 6.65 (m$_c$, 4H, H-4), 6.95–7.12 (m, 16H, H-2/H-3) ppm. $^{13}$C-NMR (100 MHz, CDCN): $\delta$ = 119.3 (C-4), 120.0 (C-2)*, 129.5 (C-3)*, 158.6 (C-1) ppm. MS (APCI) for $C_{24}H_{20}BO_4^{-}$ [M⁻] calculated: 383.1460, found 383.1451. IR (ATR): $\tilde{\nu}/cm^{-1}$ = 3037 (w), 1592 (s), 1488 (s), 1437 (w), 1227 (s), 1155 (w), 1094 (w), 1026 (m), 1008 (m), 957 (s), 878 (m), 770 (w), 744 (s), 715 (w), 691 (s), 668 (w).

### Biphenol borate polymer (BPB-Poly)

An oven-dried Schlenk tube equipped with a stirrer and septum was charged with 4,4′-biphenol (745 mg, 4.00 mmol, 2.00 eq.), evacuated, backfilled with argon for a total of three times, and dissolved in anhydrous THF (30 ml). The reaction was started by the addition of LiBH₄ (2.00 M in anhydr. THF, 1 ml, 2 mmol, 1.00 eq.). The colourless suspension was stirred for 6 h at room temperature under an argon atmosphere. The colourless solid was collected through filtration under an argon atmosphere, and washed with anhydrous THF (4 × 10 ml), and dried under reduced pressure overnight at 100 °C. The polymer was obtained as an off-white solid (93%, 695 mg).

### Procedure for the synthesis of biphenol borate-COFs (BPB-COF)

An oven-dried Schlenk tube equipped with a stirrer and septum was charged with 4.4′-biphenol (745 mg, 4 mmol, 2.00 eq.), evacuated and backfilled with argon a total of three times, and dissolved in anhydrous THF (30 ml). The reaction was started by the addition of LiBH₄ (2 M in anhydrous THF, 1 ml, 2 mmol, 1.00 eq.) and stirred for 6 h at room temperature. The suspension was transferred into a PTFE-lined steel autoclave under an argon atmosphere. After the addition of triethylamine (192 mg, 1.9 mmol, 0.265 ml, 1 eq.) and stirring by hand, the autoclave was sealed and placed into a pre-heated oven at 150 °C for 2–12 days. After cooling down to room temperature inside the oven, the solid was collected under an argon atmosphere by filtration and

washed with anhydrous THF (4 × 10 ml). The solid was dried overnight under reduced pressure at 100 °C overnight. BPB-COF was obtained as a colourless solid (652 mg, 85%).

### Exfoliation of BPB-COF to nanosheets (BPB-NS)

For the exfoliation, 1 mg of BPB-COF (12 days), was suspended in 10 ml of dry methanol under an inert atmosphere. The suspension was sonicated for 30 min followed by centrifugation at 10,280×g for 10 min leaving the nanosheets natant in solution.

### Atomic force microscopy (AFM)

AFM was performed on a Cypher AFM Microscope (Oxford Instruments) in AC mode using OMCL-AC160TS cantilever. Si(100) wafer were sonicated for 15 min before rinsing with MilliQ H₂O. The cleaned substrates were treated for 20 min with an UV/Ozone cleaner (Ossila Ltd.) resulting in an SiO₂ substrate surface. The natant BPB-NS were drop cast on the substrate and the methanol evaporated using a light N₂ stream.

### Electrochemical impedance spectroscopy (EIS)

The powder of BPB-Poly or BPB-COF and polyvinylidene fluoride (Kynar) binder were mixed by a mass ratio of 9:1 in an Argon-filled glovebox (H₂O < 0.1 ppm, O₂ <0.1 ppm). Then, an appropriate amount of anhydrous N-methyl-2-pyrrolidone (NMP, Alfa Aesar) solvent was added to make slurry. The slurry was casted on a carbon-coated aluminum foil by a doctor blade. Subsequently, the slurry was placed on a 45 °C hot plate and dried in the glovebox overnight. Then, the temperature of the hot plate was increased to 90 °C for 6 h to remove residual NMP solvent. The electrode was punched to a circle shape with 14 mm diameter and the area of the electrode was around 1.54 cm². A 2032-type coin cell was made by placing the COF electrode between two stainless steel spacers for electrochemical impedance test. Anhydrous propylene carbonate (PC, Sigma Aldrich) was added to COF electrode before the coin cell assembly. The electrochemical impedance spectroscopy was carried out using Bio-Logic VMP3 potentiostat. The electrochemical impedance measurements were made using a 10 mV RMS potential with zero bias from the open circuit potential from 1 MHz to 10 mHz. Impedance spectra were fit using equivalent circuit modeling EC-Lab (Bio-Logic) software. The temperature dependent impedance test was carried out at 20, 30, 40, 50, and 60 °C by placing the coin cell into a digital oven and resting for at least 1 h before test. The Nyquist plot was fitted to an equivalent circuit to extract the resistivity ($R_1$) of the COF. The average thickness ($t$) of COF electrode was measured by scanning electron microscopy (SEM, FEI Nova NanoSEM230) cross-section in 5 different regions by using 7 kV accelerating voltage. The gold nano particles were deposited on the surface of electrode by a gold sputtering machine to improve electrical conductivity for good image quality. Finally, the ionic conductivity of COF was calculated by using the equation $\sigma = t R_1 \times A$, where $A$ is the area of electrode (1.54 cm²), $t$ is the thickness of electrode, and $R_1$ is extracted from the equivalent circuit fitting.

## Data availability

The data that support the findings of this study are available in the article and Supplementary Information file. Source Data are provided with this paper. Additional data are available from the corresponding author upon request. Source data are provided with this paper.

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

## Acknowledgements
Funded by the Deutsche Forschungsgemeinschaft (DFG, German Research Foundation) under Germany's Excellence Strategy—EXC 2008-390540038—UniSysCat and TH 1463/21-1. We furthermore thank the BMBF for its support (Fördermaßnahme Batterie 2020, Förderkennzeichen: 03XP0410, Dialysorb). We thank Dr. Frank Hoffmann and Prof. Michael Fröba for their advice and assessment of the structural model. Christina Eichenauer is acknowledged for the sorption measurements. Maria Unterweger is acknowledged for the PXRD measurements. Jana Lutzki is acknowledged for AFM measurements. Islam E. Khalil is acknowledged for their help with SEM imaging. Harald Link is acknowledged for ICP measurements. H.K. thanks the Einstein Center of Catalysis/Berlin International Graduate School of Natural Sciences and Engineering and acknowledges support by the IMPRS for Elementary Processes in Physical Chemistry.

## Author contributions
D.A., J.G., and A.T. were responsible for the overall direction and design of the project. D.A. performed the experimental work and interpreted the analytical data. D.A. and J.R. performed structure simulation. H.K. recorded the SEM images and S.G. the TEM images. V.C. and A.L. performed the ssNMR measurements. Y.L., S.F.L., and B.D. investigated the ionic conductivity. M.G. enabled the AFM measurements.

## Funding

## Competing interests
The authors declare no competing interests.
