## [Peer Review file · Nature Communications]

REVIEWER COMMENTS

Reviewer #1 (Remarks to the Author):

Recommendation: Too preliminary for publication.

Comments:

This manuscript from Asgari and coworkers present a 2D anionic covalent organic framework based on a tetratopic borate linkages, whose layers are connected by ionic interactions of the counteranions. The authors conducted detailed experiments to investigate the synthesis process of anionic COF (BPB-COF), structure resolution, morphological changes during crystallization, and lithium ion transport properties. However, this article lacks a certain degree of innovativeness and the experimental content seems to be too thin to adequately verify the conclusions presented in the article. Therefore, this work is currently too preliminary to be considered.

1. In fact, Zhang et al. have reported borate-centered anionic COFs (ICOFs) and applied them to Li-ion conductors in 2016 (DOI:10.1002/anie.201509014). ICOFs exhibit high specific surface area, crystallinity, ionic conductivity, and ion mobility number. Although the synthesis method of BPB-COF is inconsistent with ICOFs, this article appears to lack innovation and competitiveness in comparison.
2. The authors have tried to discuss the ionic interactions between the layers as the core innovation point of the article. I like this, but the current results of the study are not sufficient. What is the ratio of the contribution of ionic interactions and π - π interactions to interlayer stacking? A computational rationalization of who is the dominant mode of action is necessary. In addition, is this ionic interaction applicable to other anionic COFs? A generalization study is necessary.
3. What was the authors' purpose in studying the morphological changes during the crystallization of COFs by SEM and TEM? Is it to show that ionic interactions between layers guide COF crystal growth? This aspect of the study and discussion does not seem to be mentioned.
4. The study of Li-ion transport properties of BPB-COF is too simple. The idea that high crystallinity favors lithium ion transport has been confirmed by many studies, so this is not an attractive enough conclusion. Adding more information about the electrochemical properties of lithium-ion conductors and studies of interlayer ionic interactions on lithium-ion transport mechanisms are necessary.

Reviewer #2 (Remarks to the Author):

Asgari et al. reported the synthesis of a two-dimensional anionic covalent organic framework (COF) consisting of tetratopic borate linkages, with adjacent layers connected via ionic interactions. The crystalline structure was obtained by transforming an amorphous borate-linked polymer through solvothermal treatment, resulting in hollow crystallite spheres. Moreover, these crystallites were further exfoliated into nanosheets, which exhibited a tenfold increase in ion conductivity compared to the amorphous polymer, presumably due to the ordered arrangement of lithium ions within the layers. This is an interesting work, but the authors need to address the following issues before this work can be subjected to further consideration.

1. For materials synthesis, the authors mentioned “crystallized through solvothermal treatment in basic media by the addition of a small amount triethylamine.”, with 1eq of TEA shown in the figure 1b. Does the loading of the TEA affect the recrystallization of the materials? Is a catalytic amount of TEA used?
2. What role does TEA play here? Would other bases function the same?
3. For the synthesis of BPB-Poly. The authors described the product has a blue color. Is there any explanation of the color? Usually for phenol-type of structure, after oxidation, it usually turns brown or darker.
4. In the SFig1, it seems like there are multiple carbons in the structure. It would be helpful if the authors could assign those peaks.
5. In the SFig 5, the authors concluded that the structure is not stable in methanol. However, the authors exfoliated the materials to obtain the nanosheet in the methanol. Such inconsistency needs to be addressed.
6. Kind of related to the above point, the authors also characterized the material by AFM. However the morphology doesn't look similar to the image in TEM with a clear square shape. Is this due to the structural damage in methanol?
7. In PXRD, is there any explanation of the peak at 5 degree?
8. The authors should label Miller Index on the XRD figures.
9. Is there any evidence supporting the intercalation of Li-ion between adjacent layers or the Li-O bonding?
10. The authors mentioned “Xxx (Fig. 2g), which is in very good accordance to the value of 0.621 nm calculated based on the simulated model using the Pore Analyzer tool within Mercury.” But this is not the common practice to estimate the pore size for MOFs or even COFs.
11. If the lithium ion is intercalated between adjacent layers as the authors proposed, wouldn't that slow down the Li-ion conduction compared with amorphous polymer?
12. The authors claimed a 10x increase of COF sample with a conductivity of $3.1 \pm 0.3 \times 10^{-5}$ S/cm. However, such conductivity is fairly common or even below the polymer-based Li-ion conductor. Considering its low Li-ion loading, it is not really practical to use the material in the li-ion battery.

Reviewer #3 (Remarks to the Author):

Asgari et al. showed an interesting strategy for making borate COFs from borate polymers. The concept is unique. However, the authors did not show key evidence for the main findings and made big logical leaps where detailed and thorough explanations and investigations are required. Thus, this reviewer does not support publication in Nat Commun. It is more suitable for a specialized journal.

1. What is the main driving force of crystallization? Can you provide a thorough discussion of the crystallization process?
2. The BPB-COF 12 days have five main peaks (Fig. 2a). Can you indicate those five peaks to facets of the crystals?
3. Can you provide experimental evidence for the following part “The coordination of the counterion by the phenoxy groups of the borate was also observed in molecular tetraphenoxyborates and even found to act as a structural support”? Especially for the part where Li ions are working as structural supports for COF layers.
4. Why did the surface area of BPB-COFs with days of crystallization decrease and then increase, like 809 m²g⁻¹, 786 m²g⁻¹ and 825 m²g⁻¹ after 2, 5 and 12 days?
5. The following sentence does not accord with the surface area data. “At higher partial pressures the sorption isotherms of the BPB-COF prepared at longer reaction times display a steep increase in nitrogen adsorption and a hysteresis which can be attributed to the additional meso/macroporosity created through the interstices of the agglomerated crystallites (Fig. 3).”
6. The authors did not provide enough evidence, especially experimental, that Li ions are pillaring the COF layers. But then they moved to the next level of experiments assuming that sonication can remove Li ions to result in exfoliation of the layers. That is a big leap in the logic, thus problematic for the manuscript.
7. We can assume the authors’ leap is reasonable. Then, the result, exfoliation, caused by sonication could be simply because the layers are separated. The authors again showed the big leap that sonication removed Li-ion without experimental proof. Regarding this, the authors need to show whether the nanosheets contain Li ions.

Minor ones

- The referencing in the captions of Figure 1 are incorrect.

- Too many typos. I list some of them.

Sometimes “countercations”. Some other places “counter cations”

“synthetized”  “synthesized”

“though”  “through”

Many grammatical errors regarding the use of singular/plural nouns, articles, etc.

Response To Reviewers

Reviewer(s)' Comments to Author:

Reviewer #1

This manuscript from Asgari and coworkers present a 2D anionic covalent organic framework based on a tetratopic borate linkages, whose layers are connected by ionic interactions of the counteranions. The authors conducted detailed experiments to investigate the synthesis process of anionic COF (BPB-COF), structure resolution, morphological changes during crystallization, and lithium ion transport properties. However, this article lacks a certain degree of innovativeness and the experimental content seems to be too thin to adequately verify the conclusions presented in the article. Therefore, this work is currently too preliminary to be considered.

First of all, we would like to thank the reviewer for his/her efforts and expert opinion on our article. Of course, we do not entirely agree with the reviewer's opinion and hope that we can convince her/him of the quality of our work with our answers to his criticism and questions.

1. In fact, Zhang et al. have reported borate-centered anionic COFs (ICOFs) and applied them to Li-ion conductors in 2016 (DOI:10.1002/anie.201509014). ICOFs exhibit high specific surface area, crystallinity, ionic conductivity, and ion mobility number. Although the synthesis method of BPB-COF is inconsistent with ICOFs, this article appears to lack innovation and competitiveness in comparison.

We thank the reviewer for this comment. The pioneering work of Wei Zhang indeed showed for the first time that an anionic organic framework can be synthesized using anionic spiroborate centers. This work is therefore also cited prominently in our manuscript. However, while both materials utilize tetrasubstituted borates as anionic nodes there are some key differences: We demonstrate for the first time a tetratopic anionic borate linkage which is topologically (and chemically) different from the spiroborate linkage. The spiroborate linkage can only be utilized as a linear node (limiting the accessible structures and requiring chelating linkers) while the tetratopic borate node is a four coordinated center. Additionally, here phenoxy linkers are utilized for the first time, not relying on the so far reported catecholate or alcoholate containing linkers which further enrich the possible linkers and structures. As an example: In the work of Zhang et al. the linker was prepared over 5 synthetic steps while here simple commercially available 4,4'-biphenol can be used which is not suitable for the spiroborate linkage. As a consequence, the material can be prepared with ease in bigger amounts which we demonstrated by preparing 1.2 g in one batch as shown in the manuscript (compared to the 0.095 g per batch through conventional synthesis by Zhang et. al.). Please note also, that in the work of Zhang et al. the structure of the framework and structural relevancy of the counter-ion was not discussed and no simulation offered neither in the manuscript or supplementary information, that is, the crystal structure of their ICOF was actually not solved. In contrast, we provide a crystal structure refinement and show that the counter-ion is of critical importance for the final structure. Due to all of these points we believe that our work holds enough originality, significance and novelty to be considered to be published in Nat. Commun.

2. The authors have tried to discuss the ionic interactions between the layers as the core innovation point of the article. I like this, but the current results of the study are not sufficient. What is the ratio of the contribution of ionic interactions and π - π interactions to interlayer stacking? A computational rationalization of who is the dominant mode of action is necessary. In addition, is this ionic interaction applicable to other anionic COFs? A generalization study is necessary.

We thank the reviewer for these interesting questions. The first question can be probably answered even without any computational rationalization. Coulombic interactions are strong ionic interaction, while π - π -interactions are rather weak, so we can rightly assume that ionic interactions are the dominant force to determine the interlayer stacking. This is supported by the interlayer stacking distance which is with 5.016 Å much larger than graphitic stacking or usually stacking distances in 2D COFs. We do not assume that π - π -interactions play a significant role at these distances.

A generalization of such systems and interactions seems just feasible in structurally comparable systems, as the specific atomic interactions and geometries are relevant for such pillaring. As an example, the bent geometry of the biphenyl allows for the tetrahedral borate to adopt a square lattice net, whereas if the nodes would be connected by "true" linear linkers, a diamond-based net with no interlayer would form. However, as this ionic interaction was also described for discrete molecular borates, it seems feasible that there are other tetratopic borate-based frameworks capable of adopting this structure.

A sentence has been added to comment on the dominant role of ionic interactions:

"The interlayer distance of 5.016 Å is also found to be much larger than common values for two-dimensional COFs and outside the range for significant contributions through π - π interactions."

3. What was the authors' purpose in studying the morphological changes during the crystallization of COFs by SEM and TEM? Is it to show that ionic interactions between layers guide COF crystal growth? This aspect of the study and discussion does not seem to be mentioned.

We thank the reviewer for the comment. First of all, we believe that not only the changes in crystallinity or porosity during the reaction are of interest, but also the evolution of morphology, so time-dependent SEM and TEM studies are always worth measuring. In this case, however we had indeed an additional motivation to study morphological changes, which is now also described in the manuscript. During the course of the reaction, we observed that the nitrogen sorption isotherms developed a hysteresis at higher partial pressures. We assumed that the cause of the emerging hysteresis can be found in the particle morphology. Indeed, it can be seen that the hysteresis increases with the growth of crystallites located on the surface of the spheres, showing that intercrystallite voids are responsible for it.

The following has been added to the manuscript to clarify our motivation:

“For partial pressures $>0.5 p/p_0$ (Fig. 2f) an increasing hysteresis loop was observed for more crystalline samples. Through investigation with scanning and transmission electron microscopy (SEM and TEM), the emerging hysteresis was found to be consistent with crystallite length and can be attributed to the additional meso/macroporosity created through the interstices of the agglomerated crystallites (Fig. 3).”

4. *The study of Li-ion transport properties of BPB-COF is too simple. The idea that high crystallinity favors lithium-ion transport has been confirmed by many studies, so this is not an attractive enough conclusion. Adding more information about the electrochemical properties of lithium-ion conductors and studies of interlayer ionic interactions on lithium-ion transport mechanisms are necessary.*

We thank the reviewer for the comment. It is true that there have been previous reports demonstrating the higher conductivity in crystalline systems. This work, however, is focused on the development of a two-dimensional anionic covalent organic framework and not on the development of ion transport properties in this system. The studies mentioned by the reviewer regarding the interlayer ion transport are definitely interesting but beyond the scope of the current paper.

Reviewer #2

Asgari et al. reported the synthesis of a two-dimensional anionic covalent organic framework (COF) consisting of tetratopic borate linkages, with adjacent layers connected via ionic interactions. The crystalline structure was obtained by transforming an amorphous borate-linked polymer through solvothermal treatment, resulting in hollow crystallite spheres. Moreover, these crystallites were further exfoliated into nanosheets, which exhibited a tenfold increase in ion conductivity compared to the amorphous polymer, presumably due to the ordered arrangement of lithium ions within the layers. This is an interesting work, but the authors need to address the following issues before this work can be subjected to further consideration.

We thank the reviewer for this assessment.

1. *For materials synthesis, the authors mentioned “crystallized through solvothermal treatment in basic media by the addition of a small amount triethylamine.”, with 1 eq of TEA shown in the figure 1b. Does the loading of the TEA affect the recrystallization of the materials? Is a catalytic amount of TEA used?*

We thank the reviewer for the comment. We have clarified the sentence in the manuscript. A catalytic amount of TEA was not used but an equimolar amount in respect to the borate content. While lowering the amount of TEA negatively impacts the crystallization doubling the TEA loading doesn't offer any improvement. We tried to clarify the amount and role of TEA in the revised manuscript (see below, question 2)

2. *What role does TEA play here? Would other bases function the same?*

We assume that TEA does serve as a competitive nucleophile, attacking the borate center and assisting with the crystallization through an error-correction mechanism according to dynamic covalent chemistry i.e. repeated bond cleavage and formation. Additionally, possible residual phenol groups are deprotonated to facilitate the complete polymerization. Other bases were investigated for the crystallization such as lithium methoxide. The problem with LiOMe is that due to the strong basicity the polymer is disintegrated partially and the lithium-ion supplied from the LiOMe was found to replace the borate – effectively destroying the borate-COF and yielding a lithium coordination polymer.

The following has been added to the manuscript to clarify the role of TEA:

“Therefore, the resulting polymer-solvent mixture was transferred to a Teflon-lined steel autoclave and 1 equivalent triethylamine added to the suspension to deprotonate any residual phenol groups and assist as a competing nucleophile in the cleavage and reformation of the B–O bond.”

3. For the synthesis of BPB-Poly. The authors described the product has a blue color. Is there any explanation of the color? Usually for phenol-type of structure, after oxidation, it usually turns brown or darker.

We thank the author for the comment and have clarified the statement. The color of BPB-Poly is described in the study as “light blue” color but perhaps this is misleading to the reader as it is white with a very faint blue/grey tone. It is changed to off-white to not mislead the reader.

The following has been changed to not mislead the reader: “The polymer was obtained as off-white solid”

4. In the SFig1, it seems like there are multiple carbons in the structure. It would be helpful if the authors could assign those peaks.

We thank the reviewer for the suggestion. We have assigned the peaks and added the structure of the linker (Supplementary Fig. 1).

5. In the SFig 5, the authors concluded that the structure is not stable in methanol. However, the authors exfoliated the materials to obtain the nanosheet in the methanol. Such inconsistency needs to be addressed.

We thank the reviewer for the suggestion. The sentence has been specified to structural integrity, reading now: “The ability of BPB-COF to retain its structural integrity was tested in dry solvents by suspending the powder under inert atmosphere and left standing over night before filtering and drying at 100 °C.”

6. Kind of related to the above point, the authors also characterized the material by AFM. However, the morphology doesn't look similar to the image in TEM with a clear square shape. Is this due to the structural damage in methanol?

We thank the reviewer for this important question. As the AFM was not recorded in higher magnification, the morphology looks indeed different on first glance – please also note that the edges of multiple layers are harder to differentiate in AFM than TEM. We have added a magnified side-by-side view of the TEM and AFM to underline the similarity (Supplementary Fig. 23). An additional factor to consider: The AFM is recorded under ambient conditions and hence exposed to ambient air and moisture which is probably the leading reason for starting morphological damage. While the damage couldn't be avoided, we tried to minimize it as much as possible by preparing fresh samples, which were directly measured after exfoliation.

7. In PXRD, is there any explanation of the peak at 5 degree?

Yes, indeed, it is part of the framework. The Bragg reflection is present in the simulated structure. We have attached a file of the simulated structure and the observed Bragg reflections and a reflection is indeed shown at the exact 2theta degree position. (The Bragg reflection is also shown in the main graph Figure 2 b).

8. The authors should label Miller Index on the XRD figures.

We thank the reviewer for the suggestion. We have added the HKL values above the reflexes (Supplementary Fig. 4)

9. Is there any evidence supporting the intercalation of Li-ion between adjacent layers or the Li-O bonding?

We think there are several. First of all, the experimentally collected XRD pattern results in one possible structural resolution only, which is presented in the manuscript. Without intercalation of Li-ions, this structure would be hardly possible. Any other solution does not provide a good fitting structure solution. Second, the possibility for exfoliation is another proof, as a three-dimensional structure without the interlayer lithium-ions would not be able to exfoliate in a controlled fashion. Finally, a complexation of the cation by the phenoxy group as described in our system has been reported for molecular discrete systems as cited in the manuscript (*Eur. J. Inorg. Chem.* **2006**, 1690–1697; *J. Chem. Soc. Dalton Trans.* **0**, 3100–3105 see below).

To further prove the position of the Li-cations, we however also performed additional DFT simulations to obtain theoretical isotropic chemical shifts for the lithium ion based on the crystal structure, which was found to be in very good agreement with the experimental value (1.19 vs. 1.09 ppm).

This new finding has been added to the manuscript as follows: "To further validate the environment of the lithium ions present in BPB-COF, we conducted first principles quantum mechanical NMR calculations to determine the theoretical chemical shielding. Using CASTEP, the isotropic chemical shift for the lithium ion of BPB-COF was calculated to be 1.19 ppm (Supplementary Fig. 26).^{21,22} This is in accordance with the experimentally measured value of 1.09 ppm further supporting the tetrahedral coordination in the interlayer space."

10. The authors mentioned "Xxx (Fig. 2g), which is in very good accordance to the value of 0.621 nm calculated based on the simulated model using the Pore Analyzer tool within Mercury." But this is not the common practice to estimate the pore size for MOFs or even COFs.

We thank the author for the comment. The statement has been rephrased in the manuscript as follows: "Using the BETSI tool, surface areas of 802 m²g⁻¹, 784 m²g⁻¹ and 819 m²g⁻¹ (Supplementary Fig. 12) after 2, 5 and 12 days of solvothermal treatment were calculated with a pore size distribution centering around 0.61–0.64 nm (Fig. 2g, Supplementary Fig. 10)."

It is true that this is not the common practice for MOF or COFs. The pore analyzer tool is relatively new (mid 2023) hence not a lot of reports are given using this tool. However, as the pore analyzer tool was advertised by the official CCDC we do believe there is enough justification in using this tool. <https://www.ccdc.cam.ac.uk/discover/blog/new-functionality-for-researchers-investigating-porous-materials-including-metal-organic-frameworks-mofs/>

11. If the lithium ion is intercalated between adjacent layers as the authors proposed, wouldn't that slow down the Li-ion conduction compared with amorphous polymer?

We thank the author for this comment. "Free" lithium ions located randomly within the polymer seem indeed on the first glance more mobile. However, through the crystallization ordered channels are generated (as they assemble in the interlayer space). Please note, that the activation energy for Li-transport suggests that the lithium ions do not "flow" through the material but the charge is rather transported in a hopping mechanism. Therefore, the ordered channels result in a lower activation energy and increase in conductivity as the hopping of the lithium ions is now easier.

12. The authors claimed a 10x increase of COF sample with a conductivity of 3.1±0.3 ×10⁻⁵ S/cm. However, such conductivity is fairly common or even below the polymer-based Li-ion conductor. Considering its low Li-ion loading, it is not really practical to use the material in the Li-ion battery.

We agree with the referee and have therefore also refrained from presenting our COF in a striking way as a material for Li-ion batteries. However, ionic conductivity is a materials property, that also provides a lot of insights into a material structure, thus we think that these measurements are interesting on their own right. Of course, we also hope that these first values will motivate other researchers to think about structural variations of such Borate

COFs in order to further increase the ion conductivity values to the point that they become interesting for practical applications.

Reviewer #3

Asgari et al. showed an interesting strategy for making borate COFs from borate polymers. The concept is unique. However, the authors did not show key evidence for the main findings and made big logical leaps where detailed and thorough explanations and investigations are required. Thus, this reviewer does not support publication in Nat Commun. It is more suitable for a specialized journal.

We would like to thank the reviewer for his/her efforts and expert opinion on our article, but naturally not fully agree to the second part of this comment. We hope that we can convince her/him of the quality of our work with our answers to his criticism and questions.

1. *What is the main driving force of crystallization? Can you provide a thorough discussion of the crystallization process?*

We thank the reviewer for the suggestion. We have added a discussion about the crystallization in the beginning of the manuscript. When performing the polymerization within a short time span the obtained polymer (BPB-Poly) is a kinetic product. When now utilizing the solvothermal treatment the I) high temperatures, II) high pressure, III) longer time of duration are all parameters that favor thermodynamic products. Hence, we are able to obtain the now thermodynamically favored configuration of the polymer which is crystalline.

The added discussion reads as follows: "First, the phenol groups of the linker are deprotonated irreversibly by lithium borohydride, releasing hydrogen gas as byproduct for the formation of a biphenyl-linked borate polymer (BPB-Poly). Since the polymerization occurs in a short time span, first, a kinetically controlled amorphous polymer network is generated (BPB-Poly, Fig. 1b). Following the concept of *dynamic covalent chemistry*, a transformation to the crystalline compound requires the cleavage and reformation of the B–O bonds present, enabling the conversion of the kinetic and amorphous material to the thermodynamically favored crystalline state. This can be achieved through elongated reaction times, high pressure and high temperatures. Therefore, the resulting polymer-solvent mixture was transferred to a Teflon-lined steel autoclave and 1 equivalent triethylamine added to the suspension to deprotonate any residual phenol groups and assist as a competing nucleophile in the cleavage and reformation of the B–O bond. The autoclave was sealed shut and placed in the oven at 150 °C for at least 2 days (Supplementary Table 1).

2. *The BPB-COF 12 days have five main peaks (Fig. 2a). Can you indicate those five peaks to facets of the crystals?*

We thank the reviewer for the suggestion. We have added the HKL values. (Supplementary Fig. 4)

3. *Can you provide experimental evidence for the following part "The coordination of the counterion by the phenoxy groups of the borate was also observed in molecular tetraphenoxyborates and even found to act as a structural support"? Especially for the part where Li ions are working as structural supports for COF layers.*

We thank the reviewer for the comment. To further confirm the environment of the lithium ion we have performed DFT simulations to obtain theoretical isotropic chemical shifts for the lithium ion based on the crystal structure. The values were found to be in very good agreement with the experimental chemical shift (1.19 vs. 1.09 ppm).

This new finding has been added to the manuscript as follows: "To further validate the environment of the lithium ions present in BPB-COF, we conducted first principles quantum mechanical NMR calculations to determine the theoretical chemical shielding. Using CASTEP, the isotropic chemical shift for the lithium ion of BPB-COF was calculated to be 1.19 ppm (Supplementary Fig. 26).^{21,22} This is in accordance with the experimentally measured value of 1.09 ppm further supporting the tetrahedral coordination in the interlayer space."

For the coordination of the counterion within the phenoxy sites of the molecular tetraphenoxyborates please see the following article, in which single crystal structures prove the exact coordination of the counter cation. <https://chemistry-europe.onlinelibrary.wiley.com/doi/full/10.1002/ejic.200600074>. Additionally, in this publication sodium was found to also act as structural support leading to the formation of one-dimensional chains of discrete borate molecules. <https://pubs.rsc.org/en/content/articlehtml/2000/dt/b003375h>. We have provided both figures from the publications below. As obtaining a single crystal for COFs is already an extremely challenging task specially when a new linkage is investigated, we verified the structure through a whole pattern Rietveld refinement to refine the simulated model with the experimentally collected powder pattern. Other possible structures were also investigated but were not found to be matching.

Molecular structure of triphenoxyborane (**6a**) and the tetraphenoxyborate (**6b**) determined by X-ray analysis.

A view of the structural linkage between the $[B(O_2C_6H_{10})_2]^-$ anion and sodium cations in a section of one of the infinite chains of $[(DMSO)Na\{B(O_2C_6H_{10})_2\}]_n$ (**3**). The cyclohexane rings are shown as line representations and all hydrogen atoms are omitted for clarity.

4. Why did the surface area of BPB-COFs with days of crystallization decrease and then increase, like 809 m²g⁻¹, 786 m²g⁻¹ and 825 m²g⁻¹ after 2, 5 and 12 days?

We thank the reviewer for the comment, but given the only small variations we rather think that the surface areas stay essentially constant. We have now improved the surface area calculation model by using the preferred BETSI system instead of the BET-model (Supplementary Fig. S12), yielding however very similar values. As these measurements are done on different batches there is some variance to be expected with each batch. The biggest variance is 4.9% which for a series of COFs even when synthesized under the same conditions is acceptable.

5. The following sentence does not accord with the surface area data. “At higher partial pressures the sorption isotherms of the BPB-COF prepared at longer reaction times display a steep increase in nitrogen adsorption and a hysteresis which can be attributed to the additional meso/macroporosity created through the interstices of the agglomerated crystallites (Fig. 3).”

We thank the reviewer for pointing this out, in fact this sentence was phrased unclear. Only the hysteresis and **not** the steep nitrogen uptake is impacted by the longer reaction times and therefore does not impact the surface area as the area is determined in the partial pressure range around 0.2.

The part has been revised and reads as follows: “Notably, while the elongation of the solvothermal treatment results in an improved crystallinity (Fig. 2a), no significant enhancement in surface area could be detected through N₂ sorption analysis (Fig. 2f, inset). For partial pressures >0.5 p/p₀ (Fig. 2f) an increasing hysteresis loop was observed for more crystalline samples. Through investigation with scanning and transmission electron microscopy (SEM and TEM), the emerging hysteresis was found to be consistent with crystallite length and can be attributed to the additional meso/macroporosity created through the interstices of the agglomerated crystallites (Fig. 3).”

6. The authors did not provide enough evidence, especially experimental, that Li ions are pillaring the COF layers. But then they moved to the next level of experiments assuming that sonication can remove Li ions to result in exfoliation of the layers. That is a big leap in the logic, thus problematic for the manuscript.

We thank the reviewer for the suggestion and have revised the paragraph as it seemed to be unclear. First of all, it should be stated that the exfoliation is already an experimental indication for the pillaring of the COF layers by Li ions. As already mentioned, we have now performed additional DFT simulations based on the crystal structure further confirming the lithium ion environment. We have also used the experimental PXRD data for a whole pattern Rietveld refinement which takes into account the I) Diffraction position, II) diffraction intensity and III) peak shape. In contrast to the often-encountered Pawley refinement, the Rietveld refinement is a suitable method to determine the absolute structure of a crystalline material. The next logical question and additional proof for the layered structure at hand was to break the weaker ionic interactions while retaining the covalent bonds inside the layers which we also successfully were able to demonstrate through the exfoliation. We think that taking all these points together there is enough solid evidence for the suggested structure.

The paragraph has been revised as follows: "Having the rectangular cuboid crystallites of BPB-COF, we reasoned that we can deconstruct the crystallites in a top-down approach. As the layers are comprised from covalent bonds, disruption of the weaker ionic interactions pillaring the layers should yield borate-based covalent organic nanosheets (BPB-NS). We hypothesized that in order to achieve a chemical exfoliation the solvent should have an affinity for the lithium ion in order to compete with the ionic interactions between the phenoxy moieties and the lithium ions. While ultrasonication in THF proved to be unsuccessful, the exfoliation could be achieved by using dry methanol instead. First, BPB-COF (12 days) was suspended in dry methanol followed by sonication at 35 kHz for 30 minutes. After centrifugation to settle bigger particles and fragments, the desired BPB-NS were left natant in solution (Fig. 4a)."

7. We can assume the authors' leap is reasonable. Then, the result, exfoliation, caused by sonication could be simply because the layers are separated. The authors again showed the big leap that sonication removed Li-ion without experimental proof. Regarding this, the authors need to show whether the nanosheets contain Li ions.

Upon suspension and sonication in deuterated dry ethyl alcohol we do see a ^7Li resonance expected for solvated ions. At the same time no sharp resonance in the ^{11}B spectra can be seen which does evidence that the lithium ions are removed from their phenoxy coordination site (see below). However, as this is only qualitative, as mentioned directly above we have revised this part and don't claim a removal of lithium ions but a separation of the layers. Characterization of the nanosheets unfortunately is very impractical as the elements are I) too light for EDX analysis in TEM and their thickness combined with the sensitivity of BPB-COF towards moisture and ambient conditions makes the characterization very challenging.

REVIEWERS' COMMENTS

Reviewer #1 (Remarks to the Author):

While I don't fully agree with the author's response, perhaps scientific research advances in such a process, and I agree that this article can be published in Nature Communications after formatting confirmation.

Reviewer #2 (Remarks to the Author):

It seems the framework reported in this work is the same as one of those frameworks reported in a recently published Science paper (<https://www.science.org/doi/10.1126/science.adj8791>), although this work reports ion conducting property, while the Science paper reports gas adsorption/separation property.

This referee appreciates the authors' efforts trying to address those questions and concerns. If the other two referees are satisfied with the revision, this referee is fine with its publication on Nat. Commun.

Reviewer #3 (Remarks to the Author):

The authors tried to answer the comments and did some control experiments. However, the quality of the data and depth of the discussion in the answer is not satisfactory. Especially the mechanism and conduction of Li⁺. Thus, this work is more suitable for a more specialized synthesis journal.